# Production of Genetically Modified Porcine Embryos via Lipofection of Zona-Pellucida-Intact Oocytes Using the CRISPR/Cas9 System

**DOI:** 10.3390/ani13030342

**Published:** 2023-01-18

**Authors:** Celia Piñeiro-Silva, Sergio Navarro-Serna, Ramsés Belda-Pérez, Joaquín Gadea

**Affiliations:** 1Department of Physiology, International Excellence Campus for Higher Education and Research “Campus Mare Nostrum”, University of Murcia, 30100 Murcia, Spain; 2Institute for Biomedical Research of Murcia (IMIB-Arrixaca), 30120 Murcia, Spain; 3Faculty of Bioscience and Technology for Food, Agriculture and Environment, University of Teramo, 64100 Teramo, Italy

**Keywords:** CRISPR/Cas9, lipofection, porcine oocytes, electroporation

## Abstract

**Simple Summary:**

Genetically modified pigs are very useful thanks to their applications in basic research, biomedicine, and meat production. There are different methods for producing them, including cloning and the microinjection or electroporation of oocytes and zygotes. Easier techniques are being developed, such as lipofection, which involves the encapsulation of the CRISPR/Cas9 system into vesicles that are introduced into cells. We compared the embryo development and mutation rates associated with different conditions of lipofection treatment with the electroporation technique in zona-pellucida-intact porcine oocytes. We found that the lipofection treatment, once optimized, was as effective as the electroporation technique in terms of the embryo development and mutation rates. In addition, an increment in the concentration in the media of the liposomes–CRISPR/Cas9 system complexes had a detrimental effect on the embryo development parameters, which could indicate a possible toxic effect. The achievement of generating mutant embryos via lipofection without removing the zona pellucida could open up a new, easy, and cheap way of producing genetically modified pigs.

**Abstract:**

The generation of genetically modified pigs has an important impact thanks its applications in basic research, biomedicine, and meat production. Cloning was the first technique used for this production, although easier and cheaper methods were developed, such as the microinjection, electroporation, or lipofection of oocytes and zygotes. In this study, we analyzed the production of genetically modified embryos via lipofection of zona-pellucida-intact oocytes using Lipofectamine^TM^ CRISPRMAX^TM^ Cas9 in comparison with the electroporation method. Two factors were evaluated: (i) the increment in the concentration of the lipofectamine–ribonucleoprotein complexes (LRNPC) (5% vs. 10%) and (ii) the concentration of ribonucleoprotein within the complexes (1xRNP vs. 2xRNP). We found that the increment in the concentration of the LRNPC had a detrimental effect on embryo development and a subsequent effect on the number of mutant embryos. The 5% group had a similar mutant blastocyst rate to the electroporation method (5.52% and 6.38%, respectively, *p* > 0.05). The increment in the concentration of the ribonucleoprotein inside the complexes had no effect on the blastocyst rate and mutation rate, with the mutant blastocyst rate being similar in both the 1xRNP and 2xRNP lipofection groups and the electroporation group (1.75%, 3.60%, and 3.57%, respectively, *p* > 0.05). Here, we showed that it is possible to produce knock-out embryos via lipofection of zona-pellucida-intact porcine oocytes with similar efficiencies as with electroporation, although more optimization is needed, mainly in terms of the use of more efficient vesicles for encapsulation with different compositions.

## 1. Introduction

According to the Food and Agriculture Organization of the United Nations [1], after poultry, pigs (*Sus scrofa*) are the second most commonly consumed meat source in the world. Therefore, they are an extremely important species agriculturally. Furthermore, their physiological and anatomical similarities to humans make pigs a great model for biomedical purposes (recently revised by Navarro-Serna et al. [2]). For these reasons, genetically modified pigs are produced for a variety of applications, including the investigation of human diseases [3,4], xenotransplantation research [5,6], and the improvement of animal production [7,8].

Regarding agriculture, an important issue is the prevalence of different viral diseases that can result in severe economic losses. The most important one is porcine reproductive and respiratory syndrome (PRRS) [8,9,10]. The symptoms of this syndrome in pregnant sows include anorexia, late-term abortions, weak piglets, and delayed return to estrus. In piglets, PRSS causes diarrhea, respiratory disorders, and increased preweaning mortality [11,12]. These worldwide economic losses can be avoided via the production of genetically modified pigs that are resistant to this virus. CD163 is a membrane protein located in different subtypes of macrophages that is involved in the recognition of various ligands. This protein was identified as the fusion receptor for the PRRS virus [13], and its removal in CD163-KO pigs was found to make them resistant to the PRRS virus [14].

Somatic cell nuclear transfer (SCNT) has been used as the main technique by which genome-engineered pigs are produced, whereby a genetically modified cell is used as the donor nucleus. With the development of new restriction endonucleases, the most important one being the clustered regularly interspaced short palindromic repeats (CRISPR)/CRISPR-associated gene 9 (Cas9) system, the efficiency of the generation of mutant cells for use in SCNT increased [15,16]. Furthermore, genetically modified pigs were also produced via direct modification of embryos with CRISPR/Cas9 using techniques such as microinjection [8] and electroporation [17]. However, SCNT and microinjection are difficult to perform, requiring specific equipment and trained personnel. While the electroporation technique is the easiest one to perform, an electroporator is needed, and the efficiency of generating KO porcine embryos is equivalent to that achieved via microinjection [18].

To avoid these disadvantages, lipofection may be an alternative method for producing genetically modified animals. Since its development in 1987 by Felgner et al. [19], lipofection has been a common transfection procedure used to introduce foreign molecules into cells. It involves the encapsulation of foreign molecules into liposomes formed from cationic lipids. These complexes are introduced into cells through a fusion and endocytosis process [20]. This procedure has been used successfully in many different types of porcine cells, including neural stem cells [21], epithelial cells [22], fibroblasts [23], granulosa cells [24], and even embryonic cells [25].

The CRISPR/Cas9 system can also be introduced into cells using the lipofection method, and it can be used to produce genetically modified animals. Currently, there are only a few reports from the same research group in Japan on the use of lipofection in *zona pellucida* (ZP)-free porcine oocytes and embryos [26,27,28].

The use of ZP-free oocytes/embryos comes with disadvantages regarding manipulation and viability, so the optimization of the process in ZP-intact oocytes and embryos is needed. Recently, lipofection in ZP-intact embryos has been reported, albeit without success, as no mutant embryos have been obtained [29].

For this reason, the aim of this work was to produce CD163 KO embryos via lipofection of ZP-intact oocytes, evaluate the efficiency of the method, and compare it with a standard electroporation method. Once the lipofection methodology is fully optimized, it will facilitate the generation of genetically edited embryos and animals for different models of interest in the biomedical and agriculture industries.

## 2. Materials and Methods

### 2.1. Ethical Issues

This study was developed in accordance with European Union Directive 2010/63/EU and the Spanish Policy for Animal Protection (RD 53/2013 fi). The Ethics Committee of the University of Murcia and Murcia Regional Government for the use of genetically modified organisms approved this project (reference CBE 195/2019, CCEA 525/2019; reference 01/2016, activities A/ES/16/79, facilities A/ES/16/I-22 and I-23).

### 2.2. Culture Media Reagents

All the reagents were obtained from Sigma-Aldrich Quimica, S.A. (Madrid, Spain) unless otherwise indicated.

### 2.3. Single Guide RNA (sgRNA) Design

A new single guide sequence targeting exon 7 of the *CD163* gene was designed using the software available from CNB-CSIC (https://bioinfogp.cnb.csic.es/tools/breakingcas, accessed on 10 January 2022) [30]: 5′-TACTTCAACACGACCAGAGCAGG (Figure 1). Both the sgRNAs and Cas9 protein were obtained from IDT (Integrated DNA Technologies, Coralville, IA, USA), and the RNP complex was prepared according to the manufacturer’s instructions.

### 2.4. In Vitro Maturation of Oocytes (IVM)

The cumulus–oocyte complexes (COCs) were obtained from gilt ovaries from a slaughterhouse and processed as previously described [31]. Briefly, the ovaries were transported in saline solution at 38 °C and washed once in 0.04% cetrimide solution and then in saline solution, both at 38 °C. Fluid from follicles of 3–6 mm in diameter was aspirated, and good quality COCs were selected, washed in Dulbecco’s PBS (DPBS) with 0.2 g/L polyvinyl alcohol (PVA), and then in maturation medium supplemented with 10% porcine follicular fluid (NCSU37). After washing, groups of 50 COCs were cultured in 500 µL of NCSU37 supplemented with 40 ng/mL fibroblast growth factor 2 (FGF2), 20 ng/mL leukemia inhibitory factor (LIF), and 20 ng/mL insulin-like growth factor 1 (IGF1) [32] at 38.5 °C and 5% CO_2_. During the first 20–22 h, the media were supplied with dibutyryl 1 mM cAMP, 10 IU/mL eCG, and 10 IU/mL hCG, followed by 20–22 h in NCSU37 supplemented with FGF2, LIF, and IGF1 without dibutyryl cAMP, eCG, and hCG. After IVM, the COCs were denuded of cumulus cells via the addition of 50 µL hyaluronidase at 0.5% to each well of NCSU37 and gentle pipetting until most of the cumulus cells were removed [33].

### 2.5. Lipofection Treatment

Lipofection with Lipofectamine^TM^ CRISPRMAX^TM^ Cas9 (Thermo Fisher, Waltham, MA, USA) was performed at the same time as in vitro fertilization (IVF) according to the manufacturer’s instructions.

Briefly, for the standard concentration, 12.5 μL of Opti-MEM I Reduced Serum Media (Thermo Fisher, Waltham, MA, USA) was well mixed with Cas9 protein (final concentration of 50 ng/μL), sgRNA (final concentration of 25 ng/μL), and 1.25 μL of Cas9 Plus^TM^ Reagent. The solution was incubated for 5 min at room temperature (RT). Meanwhile, 12.5 μL of Opti-MEM I Reduced Serum Media was mixed with 0.75 μL of CRISPRMAX^TM^ transfection reagent and incubated at RT for 3 min. After incubation, both solutions were well mixed and incubated at RT for 10–20 min. The resulting solution was added to each well of 500 μL medium during IVF.

### 2.6. Electroporation Treatment

The electroporation treatment was performed as previously described [34]. Briefly, after washing in Opti-MEM I Reduced Serum Media, the oocytes were electroporated in a slide between 1 mm gap electrodes (45-0104, BTX, Harvard Apparatus, Holliston, MA, USA) connected to an ECM 830 Electroporation System (BTX, Harvard Apparatus, Holliston, MA, USA) using 4 pulses of 30 V at a 1 ms pulse duration and a 100 ms pulse interval with a concentration of Cas9 protein and sgRNA of 50 ng/μL and 25 ng/μL, respectively (the same concentrations as in the lipofection treatment).

### 2.7. In Vitro Fertilization (IVF) and Embryo Culture

The procedures for IVF were mainly the same as those described in previous work [31]. In vitro matured oocytes were transferred to IVF-TALP (TALP medium [35] supplemented with 1 mM sodium pyruvate, 0.3% BSA, and 50 µg/mL gentamycin). The oocytes were inseminated with frozen-thawed ejaculated spermatozoa from a tested boar after being selected using a swim-up procedure [33]. Briefly, a 0.25 mL straw of semen was thawed in a water bath for 30 s at 38 °C. The semen was diluted in 2 mL NaturARTsPIG sperm swim-up media (Embryocloud, Murcia, Spain) at 38 °C. The quality of the semen after thawing was evaluated and the total sperm motility was found to be >60%, the vitality >80%, and the morphoanomalies <10%.

For the sperm selection, the swim-up was performed as in previous work [33]. The sperm was diluted in IVF-TALP and the oocytes were inseminated at a final concentration of 3000 sperm/mL. The gametes were cocultured at 38.5 °C, 5% CO_2_, and 7% O_2_ for 18–20 h.

### 2.8. In Vitro Embryo Culture (EC)

After co-incubation in TALP for 18–20 h, the remaining cumulus cells and zona-attached sperm were removed from the putative zygotes via pipetting. The putative zygotes were cultured in NCSU23a (containing 0.5 mM sodium pyruvate and 5 mM sodium lactate) for 24 h and then in NCSU23b (containing 5.55 mM glucose) until 156 h after fertilization at 38.5 °C, 5% CO_2_, and 7% O_2_ [31]. After the NCSU23a culturing, the cleavage was evaluated and 2–4 cell embryos were transferred to NCSU23b in a different well than that containing putative zygotes that did not divide. On day 6.5, the blastocyst formation rate was evaluated and the blastocysts were collected.

### 2.9. Mutation Analysis

The blastocysts were washed in nuclease-free water and stored individually with a minimum volume (2–5 μL) at −20 °C until analysis. Genomic DNA extraction and PCR were carried out using a Phire Animal Tissue Direct PCR Kit (Thermo Fisher, Waltham, MA, USA) according to the kit’s protocol. The 12.5 μL PCR reaction was performed with a primer concentration of 0.5 μM (forward: 5′-TTGTCTCCAGGGAAGGACAGG; reverse: 5′-AGAGTGAAAGGTGGGACTCG). The PCR cycling times were 5 min at 98 °C, followed by 40 cycles (denaturation for 5 s at 98 °C, annealing for 5 s at 64.3 °C, extension for 20 s at 72 °C) and final extension for 1 min at 72 °C.

The mutation detection on exon 7 was analyzed via the fluorescent PCR-capillary gel electrophoresis technique [33,36]. The PCR was performed using 6-FAM-labeled forward primers. After the PCR, the samples were processed as described previously [33] and the fluorescent PCR-capillary gel electrophoresis technique was performed using a GeneScan^TM^ 500 LIZ Size Standard (Applied Biosystem, Thermo Fisher, Waltham, MA, USA) and a 3500 Genetic Analyzer (Applied Biosystems, Thermo Fisher, Waltham, MA, USA). The details of the instrumental protocol were similar to those previously described [36]: capillary length: 50 cm; polymer: POP7; dye set: G5; run voltage: 19.5 kV; pre-run voltage: 15 kV; injection voltage: 1.6 kV; run time: 1330 s; pre-run time: 180 s; injection time: 15 s; data delay: 1 s; size standard: GS500 (−250) LIZ; and size-caller: SizeCaller v1.10. The results were analyzed using Gene Mapper 5 (Life Technologies, Carlsbad, CA, USA).

When the peak obtained via capillary electrophoresis was the same size as the control peak, the samples were considered to be WT, whereas other peaks of different sizes with respect to the control peak were considered to be KO. When more than two peaks were detected in a sample, it was evaluated as mosaic.

### 2.10. Statistical Analysis

All the data analyses were performed using SYSTAT version 13 (Systat Software, San Jose, CA, USA). The normality of the variables was tested using the Shapiro–Wilk test. As all the variables were not normally distributed, they were analyzed via the non-parametrical Kruskal–Wallis test. When significant differences were detected (*p* < 0.05), the values were compared via the Conover–Iman test for pairwise comparisons. 

## 3. Experimental Design

To evaluate the effect of the oocyte treatment on embryo development (cleavage and blastocyst rate), non-treated oocytes (control group) were compared with electroporated and lipofected oocytes before IVF.

To evaluate the efficiency of the lipofection method in relation to the gene edition, different conditions were tested, and the electroporation method was used as a control treatment for comparison in terms of the mutation (monoallelic and biallelic) and mosaicism rates. 

First, we evaluated the importance of the final concentration of the lipofectamine + RNP complex (LRNP complex) in the culture media, comparing 5% vs. 10% *v*/*v*, with fixed values for the Cas9 protein (50 ng/μL) and sgRNA (25 ng/μL). Four replicates were performed, and between 290 and 360 oocytes were evaluated per group.

Second, we evaluated the concentration of RNP in the lipofection complex, comparing 50 ng/μL of Cas9 protein and 25 ng/μL of sgRNA with 100 ng/μL of Cas9 protein and 50 ng/μL of sgRNA. Three replicates were performed, and between 170 and 225 oocytes were evaluated per group.

The embryo development parameters that were evaluated included the cleavage rate (embryos that achieved the 2-cell stage at day 2 post-insemination [pi] per total oocytes) and blastocyst rate (embryos that achieved the blastocyst stage at day 6 post-insemination per total oocytes).

The mutation parameters that were evaluated included the mutation rate (mutant blastocysts per total blastocysts), mosaicism (mosaic blastocysts per mutant blastocysts), and overall efficiency (mutant blastocysts per total oocytes).

### 3.1. Evaluation of Lipofectamine + RNP Complex Concentration

Four experimental groups were tested to evaluate the influence of the concentration of LRNP complexes (Table 1): control (without treatment), electroporated, lipofected 5% (5% *v*/*v* per well), and lipofected 10% (10% *v*/*v* per well). The developmental and mutation parameters were evaluated as described.

### 3.2. Evaluation of RNP Concentration

Three experimental groups were tested to evaluate the influence of the concentration of RNP (Table 2): electroporated, lipofected 1xRNP (50 ng/μL Cas9 protein and 25 ng/μL sgRNA) and lipofected 2xRNP (100 ng/μL Cas9 protein and 50 ng/μL sgRNA). The developmental and mutation parameters were evaluated as described.

## 4. Results

### 4.1. Concentration of Lipofectamine + RNP

The effect of the concentration of the LRNP complexes in the medium was evaluated. Regarding embryo development (Table 3), we observed an increase in the cleavage rate in the electroporated group compared with the other groups (*p* < 0.01); however, the rate of blastocysts per oocytes was similar for the control, electroporated, and lipofected 5% groups. Increasing the concentration of the lipofectamine + RNP complex in the medium from 5% to 10% (*v*/*v*) had a detrimental effect on embryo development, as the cleavage and blastocyst rates were significantly lower in the lipofected 10% group compared with the other groups.

Regarding the mutation parameters (Table 4), no significant differences were found between the groups in terms of the mutation rate (ranging from 18% to 33%) and the mosaicism rate (ranging from 0% to 6%) (*p* > 0.05). Mutant blastocysts were obtained in both groups of lipofected oocytes, showing that the method is effective in ZP-intact oocytes. 

Regarding the overall efficiency, which was measured as the number of blastocysts with at least one mutant allele per 100 oocytes treated, the electroporated and lipofected 5% groups were similar (6.38% vs. 5.52%, *p* > 0.05, Table 4). Increasing the concentration of lipofectamine + RNP complex in the medium from 5% to 10% *v*/*v* reduced the efficiency. Although the mutation rate was similar, the lower blastocyst rate of the lipofection 10% group reduced the efficiency compared with the other groups (*p* < 0.01). Therefore, an LRNP complex concentration of 5% (volume) was used in the subsequent experiment.

### 4.2. Concentration of RNP

As in the previous experiment, the use of electroporation increased the rate of cleavage compared to the use of lipofection (*p* < 0.01, Table 5); however, there were no significant differences in the blastocyst rates between the groups (*p* = 0.25; Table 5). The increased concentration of RNP in the liposomes had no detrimental effect on embryo development.

Regarding the mutation parameters (Table 6), the mutation rate and overall efficiency were similar among the groups, although they were considerably lower than in the previous experiment.

Remarkably, there was an absence of mosaic embryos in all the groups, although the mutation rate and the number of mutant blastocysts obtained were both low.

## 5. Discussion

Since the first genetically modified pigs were produced [37,38], easier, cheaper, and more efficient technologies have been developed to produce such animals, starting with the SCNT method and advancing now with the electroporation method. Even though electroporation is the easiest procedure, it still needs a specific instrument and involves the handling of oocytes/zygotes outside of an incubator.

For this reason, the use of lipofectamine in oocytes and embryos is starting to be applied and optimized [28]. Lipofection is an efficient method that has been widely used in many kinds of somatic cells to introduce foreign molecules, including the CRISPR/Cas9 system [39,40].

To the best of our knowledge, only one research group has reported the use of the lipofection method in porcine ZP-free oocytes and ZP-free embryos with some success [26,27,28,29]. As the ZP is a significant physical barrier and appears to reduce the effectiveness of lipofectamine, these researchers treated ZP-free oocytes and ZP-free embryos and achieved a moderate mutation rate (ranging from 8% to 57%) with a high level of mosaicism (ranging from 87.5% to 100% of mutant embryos) [27]. Later, with further optimization of the system using ZP-free embryos, they generated mutant embryos for different genes and produced genetically modified piglets with a monoallelic mutation for the MSTN gene [26].

ZP-free oocytes are considered to be more difficult to manipulate, and their viability is lower than that of ZP-intact oocytes [26,41]. For this reason, the optimization of a lipofection method for ZP-intact oocytes is a desired objective. In this regard, Takebayashi et al. previously tried to use lipofection in ZP-intact embryos, but with no success [29]. They also explored the use of lipofection in combination with electroporation, and they obtained similar results to the use of electroporation alone [29].

In the present study, we produced genetically modified embryos via lipofection of ZP-intact oocytes during IVF. In comparison with the work of Takebayashi et al., this achievement could be due to differences between the protocols used [29]. First, a different reagent was used. Even when the commercial reagents are the same, changes to the reagent preparation could improve the efficiency. Furthermore, the stage at which the lipofection treatment was performed differed. We lipofected oocytes during IVF, whereas they lipofected putative zygotes at 10 h post-insemination.

In our case, the lipofectamine + RNP complexes were able to traverse the ZP of the mature oocytes and enter the cells. It has been shown that porcine ZP has pores that change in size depending on the stage of the oocyte/embryo [42]. In the case of in vitro matured porcine oocytes, these pores can be more than 800 nm in diameter [42]. The liposomes of CRISPRMAX lipofectamine are around 350 nm in diameter [43], so we propose that they are able to pass through the pores of the ZP.

In the first experiment, we observed reduced effectiveness in the lipofected 10% group, possibly due to the conditions being toxic. As all the components of the LRNP complex in this group were twice that of the same components in the 5% group, the toxicity may be due to the lipofectamine, the CRISPR/Cas9 system, or a combination of both components in the lipofectamine + RNP complexes. No differences were found between the control, electroporated, and lipofected 5% groups, suggesting that the lower concentration had no toxic effect. The electroporated group had a higher cleavage rate than the control group, as determined previously, probably because the oocytes were activated by the electric pulses inducing an influx of calcium from the pulsing medium [18]. In previous studies concerning the electroporation of porcine oocytes, we analyzed the IVF procedure, showing that some of the oocytes presented one pronucleus, although the majority of them, and in a similar rate to the control, presented two pronuclei (male and female; data not shown). The difference in the cleavage rate could be due to the parthenote activation, although the blastocysts produced are mainly IVF blastocysts.

In the second experiment, only the concentration of the CRISPR/Cas9 system was increased, and no detrimental effect was observed in the 2× group compared with the 1× group. Taking this into account, the suggested toxicity observed in the first experiment may be due to the greater lipofectamine concentration. Hirata et al. did not find any toxic effect regarding the concentration of lipofectamine [26], although the reagent we used was different.

Regarding the mutation rates in the second experiment, we achieved a similar overall efficiency between the groups, although the rate for the lipofected 1xRNP (5%) and electroporated groups appeared to be somewhat lower than that achieved in the first experiment. The conditions used for the lipofected 1xRNP (5%) and electroporated groups were identical in both experiments. This variation could be due to different oocyte quality between the experiments, possibly due to seasonal and/or ambient temperature changes during the year [44,45] or the possible degradation of the sgRNA because of the freeze-thaw cycles that suffered due to its use in different dates. Furthermore, although not statistically significant, the mutation rate tended to increase when the concentration of RNP in the lipofection system was doubled. This has not been tested yet in embryos, but in somatic cells, it was found that a greater concentration of RNP in the complexes increased the mutation rate [46].

We found both concentrations of RNP in the complexes to be effective. The lower concentration of RNP may be preferable to reduce costs and minimize any detrimental effects on blastocyst development. On the other hand, the higher concentration of RNP may enhance the mutation rate.

It should be noted that no biallelic mutant embryos were obtained, as all the mutant embryos were monoallelic (with WT) or mosaic (with WT). This could be due to the contamination of the analyzed blastocysts with sperm DNA during the DNA extraction (as some sperm can still be attached to the ZP). In previous studies by Hirata et al., a similar result was obtained, as almost no biallelic mutations were found in the blastocysts and no mutant homozygote piglets were produced [26,27]. However, in future studies, the removal of the ZP before the mutation analysis should be performed to avoid sperm DNA contamination problems.

We observed that the production of genetically modified porcine embryos via lipofection of ZP-intact oocytes is possible and at efficiencies similar to those achieved with the more commonly used electroporation method. As only a few conditions of lipofection were tested, additional optimization of the process may further improve the efficiency of the method. The various parameters tested for the production of genetically modified embryos, including the time of lipofection, concentration of lipofection reagent, concentration of RNP, and stage of the oocyte/embryo [26,27,28,29], all achieved different mutation rates. Changes in the lipofection conditions may also increase the efficiency of the method.

Another parameter that can be changed is the permeability of the ZP using chemical treatments such as actinase E, as Namula et al. suggested when using electroporation [41]. In the field of gene therapy, several authors have evaluated different lipid carriers for transfection processes. The use of a lipid–peptide nanocomplex that consists of a mixture of lipids and peptides that complex electrostatically with nucleic acids could be a delivery system for the sgRNA and Cas protein [47] that achieves better results than lipofectamine. Furthermore, other vehicles have been studied for the transfection of CRISPR/Cas9 RNP into cells for gene targeting, such as carbon quantum dots [48].

Furthermore, more data regarding the quality of lipofected embryos should be analyzed in future work, such as the gene expression, cell number, proportion from the inner cell mass and trophectoderm of the blastocysts, and viability after embryo transfer.

## 6. Conclusions

Here, we have shown that it is possible to generate CD163 KO embryos via lipofection of ZP-intact porcine oocytes. Thus, lipofection is an alternative and convenient method for producing genetically modified embryos at mutation efficiencies similar to those achieved with the electroporation method. Further research and optimization of the lipofection conditions are needed. The application of lipofection to generate gene-edged pig embryos is an appealing option, as the technical resources and equipment required are less demanding than those for the alternative techniques of SCNT, microinjection, or electroporation.

## Figures and Tables

**Figure 1 animals-13-00342-f001:**
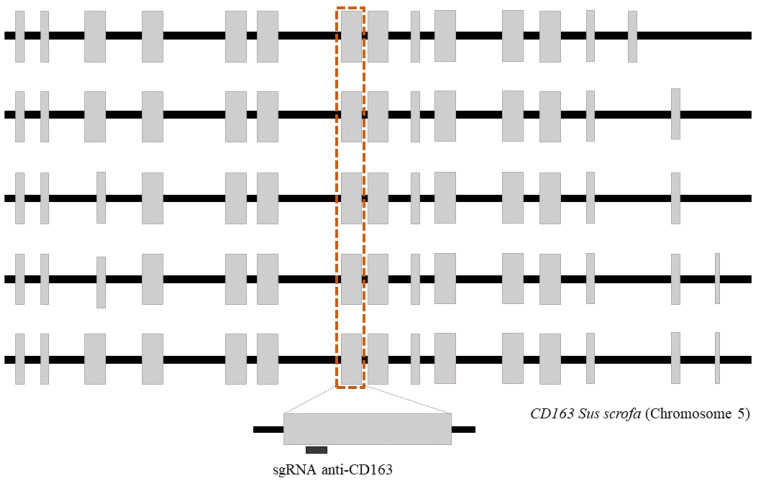
Schematic representation of *CD163* isoforms described in *Sus scrofa* (NC_010447.5). Gray boxes represent exons and black boxes represent introns. The dotted line shows exon 7, where the target region is located.

**Table 1 animals-13-00342-t001:** Experimental conditions of the different groups used to evaluate the influence of the concentration of LRNP complexes in the medium.

	Control	Lipofected 5%	Lipofected 10%	Electroporated
**Cas9 protein**	0	50 ng/uL	50 ng/uL	50 ng/uL
**sgRNA**	0	25 ng/uL	25 ng/uL	25 ng/uL
**% (*v*/*v*)/well**	0	5%	10%	

**Table 2 animals-13-00342-t002:** Experimental conditions of the different groups used to evaluate the influence of the concentration of RNP in the LRNP complexes.

	Lipofected 1xRNP	Lipofected 2xRNP	Electroporated
**Cas9 protein**	50 ng/uL	100 ng/uL	50 ng/uL
**sgRNA**	25 ng/uL	50 ng/uL	25 ng/uL
**% (*v*/*v*)/well**	5%	5%	

**Table 3 animals-13-00342-t003:** Embryo development parameters (cleavage and blastocyst formation rates) after electroporation or lipofection with the CRISPR/Cas9 system. Variables were analyzed via the non-parametrical Kruskal–Wallis test. When significant differences were detected (*p* < 0.05), the values were compared via the Conover–Iman test for pairwise comparisons.

Group	Cleavage ^1^		Blastocyst/Oocyte ^2^	
	n	%	n	%
**Control**	150/295	50.85% ^a^	61/295	20.68% ^a^
**Electroporated**	197/292	67.47% ^b^	61/292	20.89% ^a^
**Lipofected 5%**	161/348	46.26% ^a^	71/348	20.40% ^a^
**Lipofected 10%**	109/358	30.45% ^c^	48/358	13.41% ^b^
***p* value (treatment)**		<0.01		0.03

n, number of analyzed samples. ^a–c^ Values in the same column with different superscripts are significantly different (*p* < 0.05). ^1^ Two cell embryos per total number of inseminated oocytes. ^2^ Blastocysts obtained per total number of inseminated oocytes.

**Table 4 animals-13-00342-t004:** Mutation and mosaicism rates in pig embryos after electroporation or lipofection with the CRISPR/Cas9 system. Variables were analyzed via the non-parametrical Kruskal–Wallis test. When significant differences were detected (*p* < 0.05), the values were compared via the Conover–Iman test for pairwise comparisons.

Group	Mutation Rate ^1^		Mosaicism Rate ^2^		Monoallelic KO/Total ^3^		Mutant Blastocyst Rate ^4^	
	n	%	n	%	n	%	n	%
**Electroporated**	15/50	30.00%	3/50	6.00%	12/50	24.00%	15/235	6.38% ^a^
**Lipofected 5%**	16/48	33.33%	2/48	4.17%	14/48	29.17%	16/290	5.52% ^ab^
**Lipofected 10%**	6/33	18.18%	0/33	0.00%	6/33	18.18%	6/297	2.02% ^b^
***p* value (treatment)**		0.28		0.52		>0.05		<0.01

n, number of analyzed samples. ^a–b^ Values in the same column with different superscripts are significantly different (*p* < 0.05). ^1^ Percentage of embryos with 1 or more mutant alleles. ^2^ Percentage of mutant embryos with more than 2 alleles with respect to total embryos. ^3^ Percentage of mutant embryos with 1 mutant allele and 1 WT allele with respect to total embryos. ^4^ Percentage of blastocysts with at least 1 mutant allele per number of oocytes treated.

**Table 5 animals-13-00342-t005:** Embryo development parameters (cleavage and blastocyst formation rates) after electroporation or lipofection with different concentrations of the CRISPR/Cas9 system. Variables were analyzed via the non-parametrical Kruskal–Wallis test. When significant differences were detected (*p* < 0.05), the values were compared via the Conover–Iman test for pairwise comparisons.

Group	Cleavage ^1^		Blastocyst/Oocyte ^2^	
	n	%	n	%
**Control**	88/202	43.56% ^a^	30/202	14.85%
**Electroporated**	122/174	70.11% ^b^	35/174	20.11%
**Lipofected 1xRNP**	97/210	46.19% ^a^	39/210	18.57%
**Lipofected 2xRNP**	104/224	46.43% ^a^	38/224	16.96%
***p* value**		<0.01		0.57

n, number of analyzed samples. ^a–b^ Values in the same column with different superscripts are significantly different (*p* < 0.05). ^1^ Two cell embryos per total number of inseminated oocytes. ^2^ Blastocysts obtained per total number of inseminated oocytes.

**Table 6 animals-13-00342-t006:** Mutation parameters and mosaicism rate in pig embryos after electroporation or lipofection with different concentrations of the CRISPR/Cas9 system. Variables were analyzed via the non-parametrical Kruskal–Wallis test.

Group	Mutation Rate ^1^		Mosaicism ^2^		Monoallelic KO/Total ^3^		Mutant Blastocyst Rate ^4^	
	n	%	n	%	n	%	n	%
**Electroporation**	4/30	13.33%	0/30	0.00%	4/30	13.33%	4/112	3.57%
**Lipofected 1xRNP**	2/23	8.70%	0/23	0.00%	2/23	8.70%	2/114	1.75%
**Lipofected 2xRNP**	4/20	20.00%	0/20	0.00%	4/20	20.00%	4/111	3.60%
***p* value (treatment)**		0.57		1.00		0.57		0.65

n, number of analyzed samples. ^1^ Percentage of embryos with 1or more mutant alleles. ^2^ Percentage of mutant embryos with more than 2 alleles with respect to total embryos. ^3^ Percentage of mutant embryos with 1 mutant allele and 1 WT allele with respect to total embryos. ^4^ Percentage of blastocysts with at least 1 mutant allele per number of oocytes treated.

## Data Availability

The data presented in this study are available on request from the corresponding author.

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
