# Peer review of "Production of Genetically Modified Porcine Embryos via Lipofection of Zona-Pellucida-Intact Oocytes Using the CRISPR/Cas9 System"

_animals, 2023, doi:10.3390/ani13030342_

Round 1

Reviewer 1 Report

This study by Piñeiro-Silva et al. provided evidence that lipofection or porcine oocytes with intact zona pellucida can be used to produce knock-out embryos by mean of the CRISPR/Cas9 system with similar efficiency of electroporation. The lipofection approach is faster and easier to perform compared to other methods that are currently used to incorporate the CRISPR/Cas9 system into oocytes and zygotes. With further refinements to improve its overall efficiency, this technology may facilitate the production of gene edited pigs, as well as other livestock.

The experiments performed in this manuscript were well designed, data were properly interpreted, reported, and discussed.

The manuscript is overall well-written and presented, but the text should be reviewed, and some sentences could be improved.

Table 1 is unnecessary, since only 2 primers are described. This information can be moved to the text, e.g., replace “(Table 1) by “(Forward: 5’-TTGTCTCCAGGGAAGGACAGG; Reverse: 5’-AGAGTGAAAGGTGGGACTCG)” in line 183.

Author Response

Reviewer #1 wrote 
This study by Piñeiro-Silva et al. provided evidence that lipofection or porcine oocytes with intact zona pellucida can be used to produce knock-out embryos by mean of the CRISPR/Cas9 system with similar efficiency of electroporation. The lipofection approach is faster and easier to perform compared to other methods that are currently used to incorporate the CRISPR/Cas9 system into oocytes and zygotes. With further refinements to improve its overall efficiency, this technology may facilitate the production of gene edited pigs, as well as other livestock.

The experiments performed in this manuscript were well designed, data were properly interpreted, reported, and discussed.

The manuscript is overall well-written and presented, but the text should be reviewed, and some sentences could be improved.

Table 1 is unnecessary, since only 2 primers are described. This information can be moved to the text, e.g., replace “(Table 1) by “(Forward: 5’-TTGTCTCCAGGGAAGGACAGG; Reverse: 5’-AGAGTGAAAGGTGGGACTCG)” in line 183.

Answer to reviewer #1 
We appreciate the work of the reviewer for evaluating and revising this manuscript. 
Thank you for your positive comments. We agree this technology may facilitate the production of gene edited animals. 
We have revised the text to improve some sentences (see highlighted version of the manuscript). 
According to your suggestions we have deleted the table 1 and integrated the information in the text. 
We hope that new revised version could be acceptable for publication in Animals.
Thank you for your work as reviewer of this manuscript. 

Sincerely 
Celia Piñeiro-Silva 
Joaquín Gadea

Reviewer 2 Report

The authors perform a method to produce CD163 KO embryos by lipofection of ZP-intact oocytes and aim to evaluate its efficiency and comparing it with that obtained by a standard electroporation method. This is new and bring scientific soundness. However, some methodological points should be highlighted and reconsidered. 

1- Statistical analysis: The authors did not provide enough information about the response variables that will be evaluated neither citation of the used statistical software. Also, the paragraph can be rewrtitting to clarify the flow of parametric and non-parametric analyses. 

2- Experimental design: In the 213-214, the authors describe: "...with the electroporation method used as a control treatment for comparison, in terms of embryo development to the blastocyst stage, mutation and mosaicism rates." However, in the statistical tests, the treatmentes were overall compared. Therefore, I suggest that the authors define positive (eletroporaction) and negative (without treatment) controls, and then all multiple pairwise comparisions among the experimental groups. 

3- Results: Authors must describe the statistical test used in each table presented. 

The results of all experiments were expressed in rate (%). Why did not the authors use data transformation (arcsine, for example) instead of non-parametric analysis?

The "n" experimental was different in each analysis performed in this study. In Table 4, for example, the CV was 10.78% among experimental groups. In Table 6, the CV was 9.34%. I would like the authors to comment on how variations in sample size can affect the statistical results and whether the CVs are acceptable for the analyses performed in this manuscript.

Minor comments/suggestion are highlighted inr red in PDF attached file.

Author Response

Reviewer #2 wrote 
The authors perform a method to produce CD163 KO embryos by lipofection of ZP-intact oocytes and aim to evaluate its efficiency and comparing it with that obtained by a standard electroporation method. This is new and bring scientific soundness. However, some methodological points should be highlighted and reconsidered. 
Answer to reviewer #2 
We appreciate the work of the reviewer for evaluating and revising this manuscript. 
Thank you for your positive comments. 
We will try to give you answer to your specific comments and suggestions:

Reviewer #2 wrote 

1- Statistical analysis: The authors did not provide enough information about the response variables that will be evaluated neither citation of the used statistical software. Also, the paragraph can be rewrtitting to clarify the flow of parametric and non-parametric analyses. 

Response 1: 
Thank you for your comments. We have revised and updated the statistical data analysis. 
“All data analysis was performed using SYSTAT version 13 software (Systat Software, San Jose, CA). The normality of the variables was tested using the Shapiro-Wilk test. As all the variables were not normally distributed, the variables were analysed by the nonparametrical Krustal-Wallis test. When significant differences were detected (P<0.05), values were compared by the Conover-Inman test for pairwise comparisons. “

Reviewer #2 wrote 

2- Experimental design: In the 213-214, the authors describe: "...with the electroporation method used as a control treatment for comparison, in terms of embryo development to the blastocyst stage, mutation and mosaicism rates." However, in the statistical tests, the treatmentes were overall compared. Therefore, I suggest that the authors define positive (eletroporaction) and negative (without treatment) controls, and then all multiple pairwise comparisions among the experimental groups. 

Response 2: 
Thank you for your comments. Oocyte group without treatment was the control for the variables related to embryo development (cleavage and blastocyst rate) in comparison to electroporated and lipofected oocytes. On the other hand, to evaluate the efficiency of the lipofection method on gene edition, different conditions were tested, the electroporation method used as a control treatment for comparison, in terms of mutation (monoallelic and biallelic) and mosaicism rates. 
As we mentioned in the previous response, the variables were analysed by the nonparametrical Krustal-Wallis test. When significant differences were detected (P<0.05), values were compared by the Conover-Inman test for pairwise comparisons. We have modified the paragraph to clarify the experimental design. 

Reviewer #2 wrote 

3- Results: Authors must describe the statistical test used in each table presented. 

The results of all experiments were expressed in rate (%). Why did not the authors use data transformation (arcsine, for example) instead of non-parametric analysis?

The "n" experimental was different in each analysis performed in this study. In Table 4, for example, the CV was 10.78% among experimental groups. In Table 6, the CV was 9.34%. I would like the authors to comment on how variations in sample size can affect the statistical results and whether the CVs are acceptable for the analyses performed in this manuscript.

Response 3: 
Statistical test used has been described in each table. All the variables studied were not normally distributed, so we have two possibilities for the analysis of them. One is the use of non-parametric analysis as we applied. The other is an arcsine transformation to try to normalize the data and later apply an ANOVA 
parametric test. However, not always the arcsine transformation is able to get a 
normalized distribution. According to Pelea (2018) “With the use of these 
transformations, not always the accomplishment of the assumptions of parametric analysis is achieved.” 
On the other hand, non-parametric test has not this kind of limitations. According to MacKenzie (2013): “Non-parametric tests make no assumptions about the probability distribution of the population from which the underlying data are obtained. For this reason, non-parametric tests are applicable to a wider range of data than parametric tests.” 
In consequence we decided to apply a non-parametric test. 
In relation to the sample size (n) for every group, the range was 295-358 oocytes for experiment 1 (total 1283 oocytes analysed, Table 4) and 174-224 for experiment 2 (total 810 oocytes analysed, Table 6). In our opinion, the differences in the sample size pointed out by reviewer #2 it is not a problem for the statistical analysis, because the total number of oocytes evaluated is very high (more than 2100 oocytes) and the non-parametric test used take in consideration the sample size evaluated for every experimental group. 

References 
Pelea LP. How do we proceed to violations of parametric methods assumptions? or how to work with non-normal biological variables? Revista del Jardín Botánico Nacional. 
2018;39:1-12. https://www.jstor.org/stable/10.2307/26600674
MacKenzie IS. Chapter 6 - Hypothesis Testing. In: MacKenzie IS, editor. Human-computer Interaction. Boston: Morgan Kaufmann; 2013. p. 191-232.

Reviewer #2 wrote 

Minor comments/suggestion are highlighted inr red in PDF attached file.

Thank you for your work for revising the text, we have changed and updated the text according to your suggestions. 
Line 158. “Was used semen of same boar? What were the semen quality 
parameters evaluated? “ 
Response: Yes, the semen used was of the same ejaculate of one tested boar, that was cryopreserved in straws. The quality of this semen after thawing was evaluated and the motility was over 60%, the vitality over 80% and the morphoanomalies under 10%. 
This information has been included in the text. Thank you. 
We hope that new revised version could be acceptable for publication in Animals.
Thank you for your work as reviewer of this manuscript. 

Sincerely 
Celia Piñeiro-Silva 
Joaquín Gadea 

Reviewer 3 Report

For the agricultural industry, controlling infectious diseases and keeping the health of livestock are critical matters. To investigate the pathogenesis and reproduction, the techniques regarding genetically modified pigs are powerful tools, and the improvement of these technologies is extremely important. In this study, the authors evaluated the efficiency of lipofection in the gene modification of porcine eggs by using CRISPR/Cas9 system and showed a significance potentially in this field. However, they need to address some points.

1. The authors claimed that one of the novelties of this study is showing the possibility of generating CD163 KO embryos by lipofection of ZP-intact porcine oocytes. However, CD163 KO pig itself has been already reported. What is the difference or improved point in this study?

2. The authors showed only the results of the rate to develop blastocysts (and just as tables). To evaluate the quality of these blastocysts, it is better to return them to the uterus and evaluate their development. (Is it possible to get the pups?) At least, the authors should show more information regarding the formed blastocysts (morphology, gene expression, etc).

3. I could understand the sufficiency or improvement of lipofection shown in this study compared with control-methods and electroporation, because the authors showed that no significant differences were found among these methods. The authors should describe more clearly regarding the novelty or improved point by lipofection of ZP-intact oocytes compared with the previous reports.

Author Response

Reviewer #3 wrote  
For the agricultural industry, controlling infectious diseases and keeping the health of livestock are critical matters. To investigate the pathogenesis and reproduction, the techniques regarding genetically modified pigs are powerful tools, and the improvement of these technologies is extremely important. In this study, the authors evaluated the efficiency of lipofection in the gene modification of porcine eggs by using CRISPR/Cas9 system and showed a significance potentially in this field. However, they need to address some points.

Answer to reviewer #3 
We appreciate the work of the reviewer for evaluating and revising this manuscript. 
Thank you for your positive comments. 
We will try to give you answer to your specific comments and suggestions: 

Reviewer #3 wrote  

1. The authors claimed that one of the novelties of this study is showing the possibility of generating CD163 KO embryos by lipofection of ZP-intact porcine oocytes. However, CD163 KO pig itself has been already reported. What is the difference or improved point in this study?
Response 1: The improvement point in this study is the methodology employed. We were focused on the development of a methodology that facilitate the efficient generation of KO embryo pigs. We started in our lab editing pig embryos with the optimization of microinjection and later electroporation as efficient methodologies to produce different models of KO pigs (Navarro Serna et al. 2021 y 2022). 
In this study, our objective was determinate if lipofection could be optimized to become an efficient methodology for gene editing. According to the results obtained, we thought lipofection is a valuable methodology that must be optimized in the next future. In our knowledge, it is the first report that show the mutation in pig embryos after lipofection of ZP-intact oocytes. 
In relation to the gene model chosen (CD 163KO), it was based in the worldwide interest in protecting animal health. This model for resistance to PRRS viral disease is the most recognized application of the gene editing technology for the pig industry. 
The CD163 KO pigs previously reported were produced by other techniques as somatic cell nuclear transfer, electroporation, and microinjection (revised by Navarro Serna et al 2022). In this study we produced the KO embryos by lipofection of ZP-intact oocytes, this way the manipulation of the oocytes is considerably reduced as they are not subjected to electrical pulses, mechanical stress and time outside the incubator; and the technique is less demanding, as no specific qualification or equipment is needed. 
We have included a sentence to clarify the objective and applicability of this methodology. 
References 
Navarro-Serna S, Vilarino M, Park I, Gadea J, Ross PJ. Livestock Gene Editing by 
One-step Embryo Manipulation. J Equine Vet Sci. 
2020;89:103025.doi:10.1016/j.jevs.2020.103025. 
Navarro-Serna S, Hachem A, Canha-Gouveia A, Hanbashi A, Garrappa G, Lopes JS, et al. Generation of Nonmosaic, Two-Pore Channel 2 Biallelic Knockout Pigs in One Generation by CRISPR-Cas9 Microinjection Before Oocyte Insemination. CRISPR J. 
2021;4:132-46.doi:10.1089/crispr.2020.0078. 
Navarro-Serna S, Piñeiro-Silva C, Luongo C, Parrington J, Romar R, Gadea J. Effect of Aphidicolin, a Reversible Inhibitor of Eukaryotic Nuclear DNA Replication, on the Production of Genetically Modified Porcine Embryos by CRISPR/Cas9. Int J Mol Sci. 
2022a;23:2135.doi:10.3390/ijms23042135. 
Navarro-Serna S, Dehesa-Etxebeste M, Piñeiro-Silva C, Romar R, Lopes JS, López de Munaín A, et al. Generation of Calpain-3 knock-out porcine embryos by CRISPR-Cas9 electroporation and intracytoplasmic microinjection of oocytes before insemination. 
Theriogenology. 2022b;186:175-84.doi:10.1016/j.theriogenology.2022.04.012. 
Navarro-Serna S, Piñeiro-Silva C, Romar R, Parrington J, Gadea J. Generation of Gene Edited Pigs. In: Yata VK, Mohanty AK, Lichtfouse E, editors. Sustainable Agriculture 
Reviews 57: Animal Biotechnology for Livestock Production 2. Cham: Springer 
International Publishing; 2022c. p. 71-130. 

Reviewer #3 wrote  

2. The authors showed only the results of the rate to develop blastocysts (and just as tables). To evaluate the quality of these blastocysts, it is better to return them to the uterus and evaluate their development. (Is it possible to get the pups?) At least, the authors should show more information regarding the formed blastocysts (morphology, gene expression, etc).

Response 2: This study was design to check the possibility to produce KO embryos by lipofection, and to try to optimize the process by modification of methodological factors. The parameters to be evaluated in this study included the embryo development (cleavage and blastocysts stage) and mutation rate (monoallelic and biallelic) and mosaicism rate. With this information we evaluated the efficiency of the system to produce gene edited embryos. 

We agree with reviewer #3 that the application of this methodology is the generation of gene edited pigs, by embryo transfer to recipients as we did before for gene TPC2 (Navarro Serna et al. 2021). Once we obtained KO embryos, the next step will be the embryo transfer to produce gene edited pigs in the next future. The development  post-embryo transfer is a good quality test, and it should be performed as subsequent experiments to test the viability of the embryos, but it is time-consuming and it needs a great number of staff, so the first step was to make sure that the KO embryos could be produced.  

On the other hand, we also agree with reviewer #3 in his/her commentaries about the interest in show more information regarding the blastocyst. Furthermore, even when morphologically the blastocysts look of good quality (expanded and with a good inner mass), we also agree that more information can be obtained, and it can be checked in future experiments. 

Reviewer #3 wrote  

3. I could understand the sufficiency or improvement of lipofection shown in this study compared with control-methods and electroporation, because the authors showed that no significant differences were found among these methods. The authors should describe more clearly regarding the novelty or improved point by lipofection of ZP-intact oocytes compared with the previous reports.

Response 3: Even when the differences between electroporation and lipofection groups had no significance, this means that the new method is equivalent in terms of efficiency as the normally used in the labs, even when more optimization is needed as we only check a few parameters. In comparison with electroporation, this methos doesn’t need specific equipment and the manipulation of the embryos is highly reduced, as they are not subjected to electrical pulses and environment conditions outside the incubator.  

Furthermore, in comparison with other studies, we think that the fact that we do not remove the ZP is important, as the ZP-intact embryos are easier to manipulate, it is more physiological and the exposure to pronase can be toxic to the oocytes. In addition, some researchers combined electroporation and lipofection (whithout any improvement in the efficiency), which can be a waste of time and resources, as you are conducting two methods in one oocyte. 
As we mentioned before, in our knowledge, this is the first report that show the mutation in pig embryos after lipofection of ZP-intact oocytes and open a new opportunity to improve this methodology to become more efficient. 

Finally, we hope that new revised version could be acceptable for publication in Animals.

Thank you for your work as reviewer of this manuscript. 

Sincerely 
Celia Piñeiro-Silva 
Joaquin Gadea

Reviewer 4 Report

Dear authors,

you have described a really interesting approach dealing with lipofection of oocytes that could improve experimental settings. Although this is very interesting for ongoing research, I have some minor and major comments.

Minor comments:

-Ref line 110 missing

- Why to perform a KO of Cd163 in Exon 7 and not in earlier exons? ATG is located in exon 1 and sgRNA can be designed for earlier exons.

- I only can recommend to substitute the 10% follicular fluid in your maturation medium by a chemically defined medium. Many viruses are transmitted to the offspring by incubation with follicular fluid

-As you only obtained a really low amount of monoallelic targeted oocytes and no biallelic targeted oocytes, it remains unclear if this is an easier approach to generate gene-edited pigs e.g. compared to microinjection

Major comments:

I´m really sorry to say but your data are not convincing. Cleavage rates (Table 4 and 6) of your electroporated oocytes are much higher than for the control - as you also mention in the discussion lines 350-352. However, you exclude the only possible explanation for this (parthenote formation by electroporation) yourself (line 354). Therefore, your results are not comprehensive. Moreover, your cleavage and blastocyst formation rates are really low. I usually obtain more than 80-85% cleavage, 25-30% blastocyst rate after IVF. So what happened to your controls?

Moreover, if you perform such an experiment, your oocyte quality should be comparable. As you mention yourself, experiments were conducetd over the year (lines 366-368) with a strong variation of oocyte quality. This is not acceptable for such a setting.

Author Response

Reviewer #4 wrote 
Dear authors,

you have described a really interesting approach dealing with lipofection of oocytes that could improve experimental settings. Although this is very interesting for ongoing research, I have some minor and major comments.
Answer to reviewer #4 
We appreciate the work of the reviewer for evaluating and revising this manuscript. 
Thank you for your positive comments. 
We will try to give you answer to your specific comments and suggestions:

 Reviewer #4 wrote 

Minor comments:

-Ref line 110 missing

- Why to perform a KO of Cd163 in Exon 7 and not in earlier exons? ATG is located in exon 1 and sgRNA can be designed for earlier exons.

Response 1: We targeted Exon 7 because it is the responsible of the recognition of the virus and the reported resistant CD163 KO pigs are produced using a guide against this exon, but the targeting of an early exon can be a good option to produce KO embryos as well, the guides could be also checked for their efficiency. In fact, there are resistant pigs that were produced by deleting the SRCR domain 5 (the recognition region) or by modifying it (Wells et al. 2017), this way the normal function of the CD163 protein is not altered. 
References 
Wells KD, Bardot R, Whitworth KM, Trible BR, Fang Y, Mileham A, Kerrigan MA, Samuel, MS, Prather RS, Rowland RRR. Replacement of Porcine CD163 Scavenger Receptor Cysteine-Rich Domain 5 with a CD163-Like Homolog Confers Resistance of Pigs to Genotype 1 but Not Genotype 2 Porcine Reproductive and Respiratory Syndrome Virus. 
J Virol. 2017 Jan 3;91(2):e01521-16. doi: 10.1128/JVI.01521-16

Reviewer #4 wrote 

- I only can recommend to substitute the 10% follicular fluid in your maturation medium by a chemically defined medium. Many viruses are transmitted to the offspring by incubation with follicular fluid

Response 2: Thank you for your suggestion. 
In this way, we recently tried to substitute the follicular fluid in the maturation medium for a chemically defined medium (Porcine oocyte medium-POM-, with the eCG, hCG and dibutyryl cAMP supplementation during the first 20-22 hours as described in the manuscript). However, in our hands the blastocyst development rate obtained with defined culture medium was lower in comparison with our previous conditions, as it is described in this manuscript. 

Reviewer #4 wrote 

-As you only obtained a really low amount of monoallelic targeted oocytes and no biallelic targeted oocytes, it remains unclear if this is an easier approach to generate gene-edited pigs e.g. compared to microinjection

Response 3: According to the results the efficiency of the lipofection system is 
equivalent to electroporation. 
Previously, we reported that the efficiency of electroporation and microinjection is similar when both systems are optimized (Navarro Serna 2022a and 2022b). But certainly, we did not compare lipofection vs. microinjection methodologies, and we will do in the next future to validate the advantages that the lipofection could offer. 
We believe that the improvement in the use of lipofection is in the manipulation of the oocytes, as you do not need specific training or equipment, and is faster than microinjection, so the oocytes are less time out of the incubator and no mechanic stress is produced. Furthermore, even when you produce heterozygous animals, you can breed them and have in the same litter knock-out and wild-type animals, which can be beneficial in the experiments with this kind of animals.  

References 
Navarro-Serna S, Pineiro-Silva C, Luongo C, Parrington J, Romar R, Gadea J. Effect of Aphidicolin, a Reversible Inhibitor of Eukaryotic Nuclear DNA Replication, on the Production of Genetically Modified Porcine Embryos by CRISPR/Cas9. Int J Mol Sci.

2022a;23:2135.doi:10.3390/ijms23042135. 
Navarro-Serna S, Dehesa-Etxebeste M, Piñeiro-Silva C, Romar R, Lopes JS, López de Munaín A, et al. Generation of Calpain-3 knock-out porcine embryos by CRISPR-Cas9 electroporation and intracytoplasmic microinjection of oocytes before insemination. 
Theriogenology. 2022b;186:175-84.doi:10.1016/j.theriogenology.2022.04.012.

Reviewer #4 wrote 

Major comments:

I´m really sorry to say but your data are not convincing. Cleavage rates (Table 4 and 6) of your electroporated oocytes are much higher than for the control - as you also mention in the discussion lines 350-352. However, you exclude the only possible explanation for this (parthenote formation by electroporation) yourself (line 354). Therefore, your results are not comprehensive. Moreover, your cleavage and blastocyst formation rates are really low. I usually obtain more than 80-85% cleavage, 25-30% blastocyst rate after IVF. So what happened to your controls?
Response 4: 
Regarding the cleavage rate, we detected in both experiments (Tables 4 and 6) that the cleavage rate was higher in electroporated oocytes than control (no treated) or lipofected ones (p<0.01). On the other hand, we described in discussion lines 350-352 that in our previous studies with electroporation (Navarro-Serna et al 2022a and 2022 b) we have observed the same situation for electroporated oocytes vs. control or microinjected ones. 
One possible cause of this increase could be the formation of parthenotes. Although it is possible to generate parthenotes by electrical stimulation of isolated oocytes, when we evaluated the oocytes 18-20 hours after IVF, most of the cleavage embryos derived from electroporation showed both male and female pronuclei, even when a low number of them showed only one (activated oocytes). These one pronuclei activated oocytes can be the reason of the increase in the cleavage rate. Regarding the blastocysts, as the blastocyst rate is similar to the other groups, the majority of two-cell embryos keep blocked at this stage and the number of two pronuclei zygotes is high, the blastocyst 
obtained should be in the majority from IVF. 
This and other causes of the increase in cleavage rate in electroporated oocytes must be explored and confirmed in the next future in our lab. Nevertheless, it is not the main objective of this manuscript. 
Regarding the overall rates, we suppose that they are very variant between laboratories, as they depend on many factors as the medium used, the source of ovaries and semen, the breed of the animals, the ambient temperature, or the people performing the experiments as we recently reviewed (Romar et al., 2019). It is true that the rates could be a little bit low, but they remain similar between groups and replicates in the same study. 
The cause of this relative low cleavage and blastocyst rate could be related to the quality of the oocytes recovered from the ovaries from the slaughterhouse that is out our control.  

References

Romar R, Canovas S, Matas C, Gadea J, Coy P. Pig in vitro fertilization: Where are we and where do we go? Theriogenology. 2019;137:113-

21.doi:10.1016/j.theriogenology.2019.05.045. 
Navarro-Serna S, Pineiro-Silva C, Luongo C, Parrington J, Romar R, Gadea J. Effect of Aphidicolin, a Reversible Inhibitor of Eukaryotic Nuclear DNA Replication, on the Production of Genetically Modified Porcine Embryos by CRISPR/Cas9. Int J Mol Sci. 
2022a;23:2135.doi:10.3390/ijms23042135. 
Navarro-Serna S, Dehesa-Etxebeste M, Piñeiro-Silva C, Romar R, Lopes JS, López de Munaín A, et al. Generation of Calpain-3 knock-out porcine embryos by CRISPR-Cas9 electroporation and intracytoplasmic microinjection of oocytes before insemination. 
Theriogenology. 2022b;186:175-84.doi:10.1016/j.theriogenology.2022b.04.012.  
Reviewer #4 wrote 

Moreover, if you perform such an experiment, your oocyte quality should be comparable. As you mention yourself, experiments were conducetd over the year (lines 366-368) with a strong variation of oocyte quality. This is not acceptable for such a setting.

Response 5: The source of ovaries here is very limited, overall, now whit the strong restrictions after the COVID pandemic, so we should perform the experiments over a large period of time. We try to keep the same oocyte quality over the experiments, but sometimes is impossible for external reasons out of our hands. Regardless, we keep the same quality over the replicates of each experience, and we will keep trying to improve the conditions. 
In any day of work, the pool of oocytes recovered from ovaries were randomly distributed in the 4 groups of study. So, although there were differences in oocyte quality from one session of work to other, the distribution in different experimental groups was equal. 
Finally, we hope that new revised version could be acceptable for publication in Animals. 
Thank you for your work as reviewer of this manuscript. 

Sincerely 
Celia Piñeiro-Silva 
Joaquin Gadea

Round 2

Reviewer 2 Report

The authors responded satisfactorily to the highlighted points.

The 213-215 lines must be removed, since the non parametric tests were used in the statistical analyses.

Author Response

Reviewer #2 wrote  
The authors responded satisfactorily to the highlighted points.

The 213-215 lines must be removed, since the non parametric tests were used in the statistical analyses.

Answer to reviewer #2 
Sorry for the mistake, the sentence has been deleted. 
We hope that new revised version could be acceptable for publication in Animals. 
Thank you for your work as reviewer of this manuscript. 

Reviewer 3 Report

The authors answered my comments and revised the text as making the novelity of this study more clearly to show the availavility of lipofection treatment in generating gene modified pig embryo as well as electroporation treatment. Otherwise, the authors also commented that the evaluation of the quality of blastocysts treated by each method was the future work. I think this point should be included in discussion or conclusion part. 

In addition, even if the evaluation of the quality is the future work, the authors should show the morphology of the cleavaged embryos and the developed blastocyst as table 2 in ref 31.

Author Response

The authors answered my comments and revised the text as making the novelity of this study more clearly to show the availavility of lipofection treatment in generating gene modified pig embryo as well as electroporation treatment. Otherwise, the authors also commented that the evaluation of the quality of blastocysts treated by each method was the future work. I think this point should be included in discussion or conclusion part. 

In addition, even if the evaluation of the quality is the future work, the authors should show the morphology of the cleavaged embryos and the developed blastocyst as table 2 in ref 31.

Answer to reviewer #3 
Thank you to reviewer #3 for his/her evaluation process. 
We have included a sentence in discussion related to the importance of improve the evaluation of the quality of the blastocysts treated by each method in the next future. 
There are some limitations in the simultaneously evaluation of different quality 
parameters in the blastocyst. Counting of the number of total cells by nucleus staining or the evaluation of the number of cells or proportion from inner cell mass and trophectoderm in the blastocyst by differential staining (Papaioannou and Ebert, 1988; Rivera et al., 1996) are not easy compatible with evaluation of the mutation rates (DNA extraction). 
On the other hand, the evaluation of expression level of genes related to developmental competence and quality (BMP15, GDF9, OCT4, etc…) need the RNA extraction from blastocyst. Nowadays, we are optimizing a procedure to simultaneously extract RNA and DNA from the same blastocyst that solve the evaluation of gene expression and DNA mutation in the same blastocyst. We hope to get this valuable information as soon as possible. 
We have included some representative images from blastocysts (Image 1) obtained after lipofection and electroporation. We can see that the morphology is similar between groups, so we think that these images do not add any valuable information to the reader.
Image 1. A) Blastocysts obtained after a lipofection treatment. B) Blastocysts obtained after electroporation.  (Please see the attachment)

We hope that new revised version could be acceptable for publication in Animals 

Thank you for your work as reviewer of this manuscript. 

Reviewer 4 Report

Dear authors,

thanks for your reply. Actually, I acknowledge most of your answers. Of course, it will be challenging if you don´t have appropriate slaughterhouse capacities to perform this study on the short therm. I think this manuscript is acceptable for publication if you exclude the comparison of lipofection and electroporation. You can mention this in the discussion but not in the results as this is misleading. I just can tell you from my own experience that it is possible to activate almost 100% of the embryos by electroporation and the reason is only parthenote formation. I think that the results you show for electroporation are a mixture of parthenote formation and IVF. For me this is completely misleading. You obtained good results for the IVF and lipofection but you should exclude the comparison with electroporation in this setting.

Author Response

Reviewer #4 wrote 
“thanks for your reply. Actually, I acknowledge most of your answers. Of course, it will be challenging if you don´t have appropriate slaughterhouse capacities to perform this study on the short therm. I think this manuscript is acceptable for publication if you exclude the comparison of lipofection and electroporation. You can mention this in the discussion but not in the results as this is misleading. I just can tell you from my own experience that it is possible to activate almost 100% of the embryos by electroporation and the reason is only parthenote formation. I think that the results you show for electroporation are a mixture of parthenote formation and IVF. For me this is completely misleading. You obtained good results for the IVF and lipofection but you should exclude 
the comparison with electroporation in this setting.” 

Answer to reviewer #4 

Thank you to reviewer #4 for his/her evaluation process. 

We understand your concerns about the embryo development for electroporated oocytes. We agree that when you electroporate oocytes (without spermatozoa) most of them will be activated and develop a parthenote. But it is not the same when sperm are present. Recently, we have developed a preliminary study to determinate if the blastocyst derived from oocytes electroporated and in vitro fertilized are really blastocyst or parthenotes. We have used sperm from a TPC2KO male (Navarro-Serna et al., 2021), so we can evaluate the genome of the blastocyst and determinate if they have an allele from this male (blastocyst) or not (parthenote). 

We have 3 groups, the first was the control (oocytes without electroporation and IVF), and there were two electroporated oocyte groups, one with IVF (embryos) and another without IVF (parthenotes). Preliminary results with limited sample size are shown in the next tables, new replicates are now in development. 

When we evaluated the oocytes after 20-24 of the electroporation and IVF treatment, we detected and increase in the activation rate (Pronuclei formation/matured ooctytes) in electroporated groups (90.32-100%) in comparison to control group (50%) (Table 1). Both groups with IVF treatment shown similar penetration and monospermic rates (Table 1). 

When embryo development was evaluated, we confirmed that cleavage rate was higher in electroporated oocytes (83.33 and 80.56%) than control one (35.59%) in a similar pattern to previous reported studies by our group (Navarro-Serna et al., 2022a; NavarroSerna et al., 2022b) and this manuscript. The cleavage rate was similar in electroporated oocytes with and without IVF. The blastocyst rate tended to be higher also in electroporated oocytes than control one (Table 2). 

Table 1. Parameters of porcine oocyte maturation and activation of porcine after different treatments. Penetration and monospermic rate after IVF. (Please see the attachment)
Table 2. Embryo development of porcine embryos after different treatments.  (Please see the attachment)

As reviewer #4 mentioned the electroporation of oocytes (without sperm) is able to activate them and develop the parthenotes to cleavage (80.56%) or blastocyst like stage (40.28%). 

The important question to dilucidated is if the blastocyst resulting from electroporation of oocytes and IVF are embryos or parthenotes. We evaluated the genotype from blastocyst from this group and all of them have an allele from the male, in consequence 100% of them were proper blastocyst (table 2). On the other hand, in the control group most of them 85.7 % have the father allele. Up to now, we had not the possibility to evaluate the genotype of the cleavage embryos to determinate what proportion of them are proper embryos or parthenotes. 

With these preliminary results, we can say that electroporation of oocytes induces the activation of the oocytes, increase the cleavage rate (we do not know yet what proportion are proper embryos) and that most of the blastocyst (in this case %) are not parthenotes. 

As we mentioned of the previous answer to reviewer #4, although the knowledge of the effect of the electroporation of oocytes on the embryo development is a fascinating issue, it is not the main objective of this manuscript. The main objective is to evaluate the lipofection procedure as a possible alternative to other methodologies. 
We hope that new revised version could be acceptable for publication in Animals. 

References 
Navarro-Serna, S., Dehesa-Etxebeste, M., Piñeiro-Silva, C., Romar, R., Lopes, J.S., 
López de Munaín, A., Gadea, J., 2022a. Generation of Calpain-3 knock-out porcine embryos by CRISPR-Cas9 electroporation and intracytoplasmic microinjection of oocytes before insemination. Theriogenology 186, 175-184. 
https://doi.org10.1016/j.theriogenology.2022.04.012. 

Navarro-Serna, S., Hachem, A., Canha-Gouveia, A., Hanbashi, A., Garrappa, G., 
Lopes, J.S., Paris-Oller, E., Sarrias-Gil, L., Flores-Flores, C., Bassett, A., Sanchez, R., 
Bermejo-Alvarez, P., Matas, C., Romar, R., Parrington, J., Gadea, J., 2021. Generation of Nonmosaic, Two-Pore Channel 2 Biallelic Knockout Pigs in One Generation by CRISPR-Cas9 Microinjection Before Oocyte Insemination. CRISPR J 4, 132-146. https://doi.org10.1089/crispr.2020.0078. 

Navarro-Serna, S., Piñeiro-Silva, C., Luongo, C., Parrington, J., Romar, R., Gadea, J., 2022b. Effect of Aphidicolin, a Reversible Inhibitor of Eukaryotic Nuclear DNA 
Replication, on the Production of Genetically Modified Porcine Embryos by 
CRISPR/Cas9. Int J Mol Sci 23, 2135. https://doi.org10.3390/ijms23042135. 

Papaioannou, V.E., Ebert, K.M., 1988. The preimplantation pig embryo: cell number and allocation to trophectoderm and inner cell mass of the blastocyst in vivo and in vitro. Development 102, 793-803. https://doi.org10.1242/dev.102.4.793. 

Rivera, R.M., Youngs, C.R., Ford, S.P., 1996. A comparison of the number of inner cell mass and trophectoderm cells of preimplantation Meishan and Yorkshire pig embryos at similar developmental stages. J. Reprod. Fertil. 106, 111-116. 
https://doi.org10.1530/jrf.0.1060111.
